# Knowledge search, knowledge integration and enterprise breakthrough innovation under the characteristics of innovation ecosystem network: The empirical evidence from enterprises in Beijing-Tianjin-Hebei region

**Yanli Zhang[1], Dantong Wang [2]\*, Long Xu[3]**

**1** School of Economic and Management, North China Institute of Aerospace Engineering, Lang Fang, Hebei, China, **2** School of Government, Peking University, Beijing, China, **3** School of Business Administration, Hebei University of Economics and Business, Shi Jiazhuang, Hebei, China

\* 18811569010@163.com

**Data Availability Statement:** All supporting information are available from the figshare

## Abstract

Enterprises acquire heterogeneous knowledge through external knowledge search and adapt to the change of external environment, which is of great significance to enterprise breakthrough innovation. This paper takes the innovation ecosystem as the boundary of the research paradigm. Based on innovation ecosystem theory, knowledge management theory and enterprise innovation theory, this paper constructs a moderated mediation model of the enterprise knowledge search, knowledge integration and breakthrough innovation under the characteristics of innovation ecosystem network. This model is tested on the survey data of 344 technology enterprise and manufacturing industries including R&D departments in the Beijing-Tianjin-Hebei region. The research results show that: knowledge integration plays part of the intermediary role between knowledge search and enterprise breakthrough innovation; the larger the network scale, the stronger the network connection, the stronger the intermediary role of knowledge integration on the relationship between knowledge search and enterprise breakthrough innovation. The research results reveal the important role of the innovation ecosystem in enterprise breakthrough innovation. At the same time, the research on knowledge search and enterprise breakthrough innovation should consider the network characteristics of innovation ecosystem and the ability of enterprise knowledge integration.

## Introduction

In the era of economic globalization, the external competitive environment which enterprises usually face has become more and more fierce. To address this problem, many enterprises

database (DOI number 10.6084/m9.figshare. 17049323).

**Funding:** a) Yanli Zhang, 19YJC630220, the Ministry of Education of the People's Republic of China's Youth Foundation for Humanities and Social Sciences Research Project b) Yanli Zhang, 21557652D, S&T Program of Hebei c) Yanli Zhang, HB21YJ059, Social Science Foundation of Hebei Province.

**Competing interests:** The authors have declared that no competing interests exist.

focus on breakthrough innovation, take it as an important factor for obtaining sustainable competitive advantages [1]. Breakthrough innovation refers to an innovative way for enterprises to release new products, new technologies or new services to the market [2]. The Publication of "Made in China 2025" is also a strategic plan to promote breakthrough innovation for manufacturing enterprises. The prior works conduct researches on enterprise breakthrough innovation from different perspectives. Among them, researchers particularly focus on how enterprise knowledge resources affect breakthrough innovation, which is usually based on knowledge dependence theory [3]. This theory holds that knowledge resources are important strategic resources for enterprises, which can be leveraged to deal with the changing external environment and carry out breakthrough innovation. The acquisition, sharing and development of knowledge resources are important ways for enterprises. Chesbrough and Rosenbloom are the first to propose the "open innovation" theory which is based on the concept of external knowledge search [4]. This theory believes that the open search of knowledge resources from the outside is an important way for enterprises to obtain knowledge resources and carry out breakthrough innovation effectively, especially when they face the fierce external competition environment. The current researches on knowledge search mainly includes knowledge tension theory and knowledge accumulation theory. The former emphasizes that the diversity and heterogeneity of enterprise knowledge resources can enrich the knowledge base of enterprises and affect breakthrough innovation through knowledge reorganization. The latter pays more attention on in-depth search of knowledge in one certain field, which can help the enterprises to master the frontier knowledge resources and achieve breakthrough innovation.

Social network theory was first used to study the relationship between groups, and then gradually applied to the management of enterprises and organizations. One of the social network theories is the embeddedness theory, which studies the impact of cohesion and centrality in social networks on individual behavior. Social network theory emphasizes that due to the limited amount of knowledge of the innovation subject, enterprises can search for knowledge in the network to obtain various external knowledge resources, so as to obtain the key resource elements to maintain the sustainable competitive advantage of enterprises [5]. With the increasing complexity of enterprise breakthrough innovation, on the basis of social network research, the innovation ecosystem formed by multiple innovation subjects such as enterprises, scientific research institutions and universities came into being as a complex network system to explore the collaborative innovation between enterprises and external innovation network environment, which has also become a new perspective to study enterprise breakthrough innovation [6,7]. In the innovation ecosystem, enterprises and other innovation entities rely on each other and evolve collaboratively, deeply integrating knowledge resources such as information, technology, and innovation elements. By doing this, they can gather innovation resources effectively and form an innovation network relationship. The innovation ecosystem has expanded the scope of external knowledge resources for enterprises, and has become an important channel for enterprises to search for knowledge and obtain scarce resources such as external knowledge [8,9].

In the study of the relationship between knowledge search and enterprise breakthrough innovation, Katila and Ahuja are the first to divide knowledge search into knowledge search width and knowledge search depth, which can provide novel insights for the study of enterprise breakthrough innovation by matching the theory of knowledge accumulation and knowledge tension [10]. The following literatures have conducted further empirical research on the impact of knowledge search width or knowledge search depth on enterprise innovation. However, unanimous conclusion about how to better promote enterprise breakthrough innovation has not been reached [11]. In this paper, we argue that the research on the breadth or depth of

knowledge search and breakthrough innovation in enterprise should be conducted both internally and externally. Firstly, we should take consideration of external contingency factors for further researches. Because different network characteristics of innovation ecosystem have different effects on the relationship between knowledge search and breakthrough innovation, we take the former as contingency factor to verify. Secondly, the ability of enterprise to internalize and integrate knowledge integration after acquiring external knowledge resources should be considered as an important internal factor of breakthrough innovation. Based on the above analysis, we construct a model of the enterprise knowledge search, knowledge integration and breakthrough innovation under the characteristics of innovation ecosystem network, and propose that the research on knowledge search and enterprise breakthrough innovation should consider the network characteristics of the innovation ecosystem and the capability of enterprise knowledge integration, so as to provide some reference for improving the enterprise breakthrough innovation ability.

## Literature reviews and research hypothesis

### Knowledge search and enterprise breakthrough innovation

In the innovation ecosystem, besides relying on internal knowledge resources, there are also other key paths to achieve enterprise breakthrough innovation, such as search, acquisition, integration, and utilization of external knowledge [12]. Through external knowledge search, enterprise can optimize and improve their own knowledge reserves, acquire heterogeneous knowledge, break innovation constraints, and constantly adapt to changes in the external environment [13]. The knowledge search breadth can enrich the heterogeneous knowledge stock of enterprise, and the knowledge search depth can acquire frontier knowledge from the depth direction. Enterprises can improve their innovation performance through restructuring, absorbing and utilizing the external knowledge resources. After further combing and studying the existing literatures, we find that the academic community has not reached a consensus on how knowledge search affects enterprise breakthrough innovation. Some researchers believe that there is a linear relationship between the two factors, while some others consider that the relationship has the effect of "too much is not enough" [14]. Jung and Lee believe that enterprise external knowledge search and breakthrough innovation have important linear effects. This conclusion is based on their empirical research on American enterprises from 1980 to 2006 [15]. Qin Pengfei et al. argue that knowledge search has an inverted U-shaped impact on organizational innovation capabilities, which means it will weaken innovation capabilities if too high [16].

In the Innovation Ecosystem, enterprise innovative behavior is stimulated by knowledge search. On one hand, the knowledge search breadth provides enterprise with diversified and differentiated knowledge. By broadening the acquisition channels of external knowledge resources, they can be integrated with universities and scientific research institutions. On the other hand, knowledge search depth acquires frontier knowledge in the depth direction, which can accelerate the transformation and utilization of enterprise existing knowledge and external knowledge [17]. We believe that the continuous accumulation of knowledge resources acquired by enterprise from outside will ultimately enhance the ability of breakthrough innovation. Obviously, the increasing amount of acquired knowledge resources will bring some problems to enterprise, such as the complexity and cost of knowledge management. However, the capability of enterprise knowledge integration can also be improved. Therefore, there is a linear relationship between the above two factors. In view of this, we propose the following hypotheses:

Hypothesis 1a. The breadth of knowledge search has a positive impact on enterprise breakthrough innovation.

Hypothesis 1b. The depth of knowledge search has a positive impact on enterprise breakthrough innovation.

## The mediating role of knowledge integration

The concept of knowledge integration is first proposed by Henderson and Clark, which emphasizes the systematic integration, sharing and diffusion of knowledge [18]. From the perspective of knowledge integration, enterprise uses unique knowledge integration mechanisms to reconstruct and integrates acquired knowledge resources, which is manifested as the effective integration of new and old knowledge, as well as integration and application in product development [19]. It is the important prerequisite for enterprise to achieve breakthrough innovation. The integration of heterogeneous knowledge can help enterprise to update the existing knowledge base and accelerate the generation of new knowledge, which can also provide a rich source of knowledge for enterprise to carry out breakthrough innovations. Yu and Yu believe that with the improving capability of knowledge integration, external knowledge can be integrated with enterprise knowledge highly and quickly, which can finally form a new core knowledge resources of the enterprise [20].

The search of external knowledge resources can improve the enterprise knowledge reserves for breakthrough innovation. However, the searched knowledge resources which are heterogeneous and effective are scattered and disorderly, and they are also different from the existing knowledge resources of the enterprise. Therefore, these knowledge resources must be transformed through knowledge integration and fused with the current knowledge of enterprise. Garud and Nayyar believe that in order to deal with the fierce external competition environment, it is necessary for enterprise to continuously search for external knowledge resources and introduce new external knowledge to enhance the technological innovation capabilities of enterprise [21]. The new knowledge introduce from the outside should be combined with the current knowledge structure by knowledge integration mechanism, aiming to build the innovative knowledge resources for enterprise. Jones et al. take the hospital's nursing department as a sample, study the role of knowledge integration in the relationship between interdisciplinary knowledge exchange and service content of innovative nursing department, and discuss the importance of knowledge integration [22]. Motivated by this, we propose the following hypotheses:

Hypothesis 2a. Knowledge integration plays an intermediary role in the relationship between knowledge search breadth and enterprise breakthrough innovation.

Hypothesis 2b. Knowledge integration plays an intermediary role in the relationship between knowledge search depth and enterprise breakthrough innovation.

## The regulatory role of network characteristics of innovation ecosystem

In the era of economic globalization, the breakthrough innovation of enterprise is the collaborative innovation in the symbiotic evolution innovation ecosystem, which involves the participation of innovation subjects such as universities and scientific research institutions. Therefore, the network characteristics of innovation ecosystem is an important factor for enterprise to search external knowledge resources [23]. The network scale characteristics of innovation ecosystem reflect the richness of resources that enterprise can embed from external organizational relationships. Rowley et al. believe that the more the number of external contact partners is, the richer the knowledge resources obtained from the outside [24]. The strength of

network relationship in innovation ecosystem refers to the strength of the relationship among various innovation subjects in the network. The stronger the network relationship is, the deeper the knowledge diffusion in the innovation ecosystem. Reagans et al. argue that enterprise with strong network relationship establish trust relationship with external network subjects, and have more opportunities to obtain external knowledge resources [25].

The external knowledge resources obtained by enterprise through knowledge search are not only related to the way of knowledge search, but also related to the influence of the network structure characteristics of innovation ecosystem on the relationship between knowledge search and knowledge integration. Different network scales of innovation ecosystem provide different degrees of search scope for enterprise knowledge search. The larger the width of external knowledge search, the easier the differentiated knowledge resources flow and spread among innovation subjects. The stronger the network relationship among the innovation subjects in the innovation ecosystem, the easier it is for the enterprise to communicate with other innovation subjects under the rule of trust and reciprocity, and the easier it is for the enterprise to deeply excavate, absorb and utilize the knowledge in specific fields, so that the enterprise could obtain the frontier knowledge and integrate it with the existing knowledge resources effectively [26]. Based on the data of 33 global pharmaceutical enterprises from 1975 to 2014, Wang Wei et.al prove the moderating effect of network relationship strength on knowledge search width and depth from the perspective of network attributes [27]. Therefore, we propose the following hypotheses:

Hypothesis 3a. The network size of innovation ecosystem has a positive moderating effect on the relationship between knowledge search width and knowledge integration.

Hypothesis 3b. The network connection strength of innovation ecosystem has a positive moderating effect on the relationship between knowledge search depth and knowledge integration.

## The moderated mediation role of knowledge integration

According to the above hypothesis, the mediating role of knowledge integration is moderated by the network size and network connection strength of the innovation ecosystem, which has moderated mediation effect. Therefore, we propose the following hypothesis:

Hypothesis 4a. The network size of the innovation ecosystem plays a positive role in moderating the mediating role of knowledge integration in the relationship between the breadth of knowledge search and enterprise breakthrough innovation. The larger the size of the network, the stronger the mediating role of knowledge integration between the breadth of knowledge search and the enterprise breakthrough innovation.

Hypothesis 4b. The network connection strength of the innovation ecosystem plays a positive role in moderating the mediating role of knowledge integration in the relationship between knowledge search breadth and enterprise breakthrough innovation. The stronger the network connection, the stronger the mediating role of knowledge integration between knowledge search depth and enterprise breakthrough innovation.

In summary, the concept model of this paper is shown in Fig 1.

## Materials and method

### Sample selection and data collection

The sample data of this paper is collected by means of questionnaire survey. The survey time is September 2020 to March 2021. The sample enterprises are scientific and technological

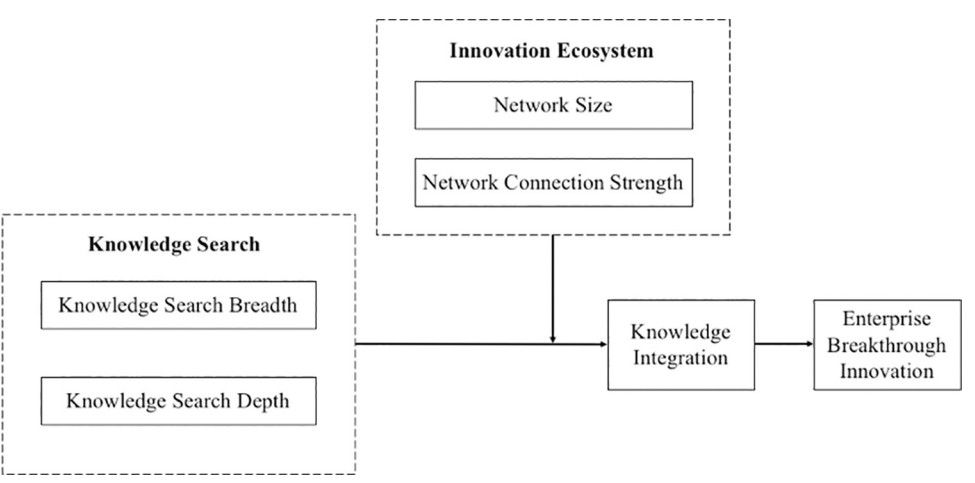

**Fig 1. Conceptual model.**

enterprises, manufacturing industry with R&D department in Beijing-Tianjin-Hebei region. These enterprises pay more attention to knowledge acquisition, integration and innovation, which is more consistent with the research theme of this paper. The questionnaires are mainly filled by enterprise managers or R&D or technical personnel, who are required to have at least 3 years working experience to ensure that they have sufficient understanding of the enterprise. This survey is divided into two stages. The first stage is the pre survey stage. In order to ensure the quality of the questionnaires, 50 enterprises are selected as samples before the formal survey. According to the suggestions of the questionnaire fillers, the choice of words and items setting of the questionnaires are revised to form the final questionnaire. The second stage is the formal survey stage. At this stage, we distribute questionnaires and collect data through three ways. The first method is to entrust University MBA students to send questionnaires to the managers or R&D or technical personnel of their enterprises through e-mail or questionnaire star network platform. The second is through communicating with government personnel, offline visiting and contacting enterprises. The third is to use personal relationships for sending electronic questionnaires to relevant enterprises by e-mail or questionnaire star network platform. A total of 500 questionnaires are distributed, including 300 paper questionnaires and 200 electronic questionnaires. A total of 378 questionnaires are collected, with a recovery rate of 75.6%. Then, we examine the returned questionnaires and eliminate the unqualified questionnaires. Finally, a total of 344 questionnaires are collected, and the effective recovery rate is 68.8%. The details are shown in Table 1.

## Variable measurement

The questionnaire of this paper is designed by 5-point Likert scale, as shown in Table 2. In the construction of the network characteristics of innovation ecosystem, we refer to the cohesion and centrality characteristics of social network theory, including two dimensions of network size and network connection strength, which have an important impact on cohesion and centrality to a certain extent. There are 10 items in the scale, the network size scale is designed with reference to the scales of Hislop [28], Xie and Zuo [29], Li et al [30]. And the network connection strength scale refers to the scales of Xie ans Zuo [29], Dodgson [31], Li et al. [30]. The knowledge search scale includes two dimensions of knowledge search depth and knowledge search width, which includes 8 items. It is designed with reference to the scales of Laursen and Salter [32], Su et al. [33]. The knowledge integration scale is designed with reference to the

**Table 1. Distribution characteristics of sample enterprises.**

| Sample characteris-tics | Category | Sam-ple Size | Propor-tion(%) | Sample characteri-stics | Category | Sam-ple Size | Propor-tion(%) |
|---|---|---|---|---|---|---|---|
| **Enterprise Scale (people)** | <100 | 128 | 37.21 | **Founding Time (year)** | <3 | 66 | 19.19 |
| | 101~300 | 69 | 20.06 | | 3~5 | 90 | 26.16 |
| | 301~500 | 58 | 16.86 | | 6~10 | 75 | 21.80 |
| | 501~1000 | 44 | 12.79 | | 11~20 | 81 | 23.55 |
| | >1000 | 45 | 13.08 | | >20 | 32 | 9.30 |
| **Current Position** | Top Managers | 81 | 23.55 | **Developm-ent Stage** | Initial Stage | 56 | 16.28 |
| | Middle Managers | 70 | 20.35 | | Growth Period | 188 | 54.65 |
| | Grassroots | 93 | 27.03 | | Mature Period | 89 | 25.87 |
| | R&D and technician | 100 | 29.07 | | Recession Period | 11 | 3.20 |
| **Nature of Enterprise** | State-Owned | 90 | 26.16 | **Location** | Beijing | 106 | 30.81 |
| | Private | 167 | 48.55 | | Tianjin | 170 | 49.42 |
| | Foreign or joint | 67 | 19.48 | | Hebei | 68 | 19.77 |

**Table 2. Variable measurement and source.**

| Variables | Questions in our questionnaire | References |
|---|---|---|
| **Network Size** | 1. There are many connections between enterprises and government.<br>2. There are many connections between enterprises and universities/scientific research institutions.<br>3. There are many connections between enterprises and intermediary organizations (or industry associations).<br>4. There are many connections between enterprises and financial institutions.<br>5. There are many connections between enterprises and peer enterprises.<br>6. There are many connections between enterprises and suppliers.<br>7. There are many connections between enterprises and customers. | Hislop [28], Xie and Zuo [29], Li et al [30]. |
| **Network Connection Strength** | 8. The enterprise has long-term cooperation and connection with other external organizations.<br>9. The enterprise has formed close cooperative relations and frequent connections with other external organizations.<br>10. The cooperation between enterprises and other external organizations has important future plans. | Xie ans Zuo [29], Dodgson [31], Li et al. [30] |
| **Breakthrough Innovation** | 1. The enterprise attaches great importance to the development of new products or services.<br>2. The enterprise can launch new products or services more quickly than peer enterprises.<br>3. The enterprise can apply breakthrough technologies to the development of new products or services. | Fey and Birkinshaw [36] and Jansen [37] |
| **Knowledge Search Breadth** | 1. The enterprise can obtain market information from suppliers and customers.<br>2. The enterprise can obtain knowledge resources from universities, government and scientific research institutions.<br>3. The enterprise can obtain knowledge resources from industry associations and intermediary organizations.<br>4. The enterprise can obtain the information on safety, technology and environmental standards of the industry. | Laursen and Salter [32], Su et al. [33] |
| **Knowledge Search Depth** | 5. The enterprise can effectively use the market information obtained from suppliers and customers.<br>6. The enterprise can effectively use the knowledge resources obtained from universities, government and scientific research institutions.<br>7. The enterprise can effectively use the knowledge resources obtained from industry associations and intermediary organizations.<br>8. The enterprise can effectively comply with the safety, technology and environmental standards of the industry. | Laursen and Salter [32], Su et al. [33] |
| **Knowledge Integration** | 9. The enterprise can systematically classify different sources and types of acquired knowledge.<br>10. The enterprise can digest and absorb external knowledge in time, which can also be mastered by individual employees.<br>11. The enterprise can integrate the acquired external knowledge into the enterprise practice and form the enterprise's knowledge system. | Cummings [34] and Zhan [35] |

scales of Cummings [34] and Zhan [35], including 3 items. The enterprise breakthrough innovation scale is designed with reference to the scales of Fey and Birkinshaw [36] and Jansen [37], including 3 items. Drawing on the existing research results of relevant literature, we select the enterprise scale (people), the funding time and the development stage as the control variables.

## Empirical analysis results

### Reliability analysis

**Exploratory factor analysis.** There are many items on the initial measurement scale of innovation ecosystem network characteristics, knowledge search, knowledge integration and enterprise breakthrough innovation, so we use CITC analysis to test whether there are useless items before exploratory factor analysis. If the CITC of the measurement item is less than 0.3, then we delete the item. The results show that the minimum CITC value of each measurement item is 0.934, so they are all retained. Then we perform KMO and Barlett test on the sample data. The KMO value is 0.928, and the Barlett test reach the significance level, which means that the data is suitable for further factor analysis.

We use the principal component analysis method for exploratory factor analysis, and use the maximum variance method for rotation. A total of 6 common factors with eigenvalues greater than 1 are extracted, and the cumulative variance contribution rate is 70.646%. The results show that all items correspond to their respective constructs and the factor loads are all above 0.5. Among them, knowledge search variables are jointly characterized by knowledge search width and knowledge search depth, and innovation ecosystem network characteristic variables are jointly characterized by network size and network connection strength.

**Trust level analysis.** Cronbach's a is used to test the reliability of the scale. The Cronbach's a coefficient values of knowledge search width, knowledge search depth, knowledge integration, enterprise breakthrough innovation, network size and network connection strength are respectively 0.844, 0.877, 0.899, 0.829, 0.730, which are all greater than the critical standard of 0.7. The results indicate that the reliability of the scale is good, as shown in Table 3.

**Validity analysis.** We test the content validity, discriminative validity and convergence validity of the scale. Since the scale uses mainly draws on the mature scales in domestic and foreign literature, and the items have been revised through the preliminary survey, the scale has good content validity. Then we use AMOS 21.0 to perform confirmatory factor analysis on variables and construct a 6-factor model. As shown in Table 4, except for the judgment standard of GFI which is slightly lower than 0.90, all other fitting indicators of the 6-factor model meet the requirements and are better than other models, indicating that the variables have good discrimination validity. At the same time, we calculate the average variation extraction (AVE) and combined reliability (CR) values of the scale. As shown in Table 3, the AVE of each variable is greater than 0.5, and the CR value is greater than 0.7, which indicates that the scale has good convergence validity.

**Table 3. Reliability and convergent validity of the scale.**

| Variable | Cronbach's a | CR | AVE |
|---|---|---|---|
| Knowledge Search Breadth | 0.844 | 0.843 | 0.573 |
| Knowledge Search Depth | 0.877 | 0.878 | 0.642 |
| Knowledge Integration | 0.899 | 0.748 | 0.899 |
| Enterprise Breakthrough Innovation | 0.829 | 0.834 | 0.628 |
| Network Size | 0.730 | 0.873 | 0.723 |
| Network Connection Strength | 0.869 | 0.891 | 0.870 |

**Table 4. Confirmatory factor analysis results of the scale.**

| Model | $\chi^2/df$ | GFI | TLI | CFI | IFI | RMSEA |
|---|---|---|---|---|---|---|
| Six Factor Model | 2.968 | 0.897 | 0.909 | 0.915 | 0.916 | 0.082 |
| Five Factor Model | 3.565 | 0.814 | 0.859 | 0.876 | 0.877 | 0.086 |
| Four Factor Model | 3.670 | 0.808 | 0.853 | 0.869 | 0.870 | 0.088 |
| Three Factor Model | 4.374 | 0.774 | 0.814 | 0.833 | 0.834 | 0.099 |
| Two Factor Model | 5.465 | 0.721 | 0.754 | 0.777 | 0.778 | 0.076 |
| Single Factor Model | 6.193 | 0.698 | 0.714 | 0.698 | 0.741 | 0.123 |

Note: 5-factor model: Combine the two dimensions of innovation ecosystem network characteristics; 4-factor model: On the basis of 5-factor model, combine the two dimensions of knowledge search; 3-factor model: On the basis of 4-factor model, combine knowledge search and knowledge integration; 2-factor model: On the basis of 3-factor model, combine innovation ecosystem network characteristics and enterprise breakthrough innovation; Single factor model: Combine all items into one factor.

**Homology analysis of variance.** In order to reduce the impact of common method variance to a certain extent, we have made certain preparations for the accuracy of item expression and anonymity in the stage of questionnaire design and data collection, and distribute questionnaires to each company. But the data obtained from the questionnaire survey conducted by a single subject will inevitably have the problem of common method variance. So we use Harman's single factor to test method for detection. From the exploratory factor analysis, we could know that the load of the first principal component is 21.385%, which means that there is no serious homology deviation problem.

## Descriptive statistical analysis

We test the average value, standard deviation and correlation coefficient of each variable in the model. As shown in Table 5, there is a significant positive correlation between knowledge search, knowledge integration and enterprise breakthrough innovation. The network size, network connection strength and enterprise breakthrough innovation also have a significant positive correlation. The results provide preliminary data support for the verification of subsequent research hypotheses and are suitable for further hypothesis testing.

## Hypothetical test

**Main effect test.** The main effects test results are shown in Table 6. Model 5 and Model 6 examine the impact of knowledge search breadth and knowledge search depth on the

**Table 5. Descriptive statistical analysis of variables.**

| Variable | Mean Value | Standard Deviation | 1 | 2 | 3 | 4 | 5 | 6 |
|---|---|---|---|---|---|---|---|---|
| Knowledge Search Breadth | 3.652 | 0.685 | 1 | | | | | |
| Knowledge Search Depth | 3.657 | 0.724 | 0.843** | 1 | | | | |
| Knowledge Integration | 3.650 | 0.750 | 0.657** | 0.771** | 1 | | | |
| Enterprise Breakthrough Innovation | 3.627 | 0.775 | 0.580** | 0.531** | 0.587** | 1 | | |
| Network Size | 3.462 | 0.544 | 0.600** | 0.629** | 0.557** | 0.428** | 1 | |
| Network Connection Strength | 3.518 | 0.768 | 0.606** | 0.577** | 0.525** | 0.402** | 0.613** | 1 |

Note: The sample size is 344

*** means p<0.001

** means p<0.01

*means p<0.05, the same below.

**Table 6. Test of main effect and mediating effect.**

| Variable | Knowledge Integration | | | Enterprise Breakthrough Innovation | | | | |
|---|---|---|---|---|---|---|---|---|
| | Model 1 | Model 2 | Model 3 | Model 4 | Model 5 | Model 6 | Model 7 | Model 8 |
| **Control Variable** | | | | | | | | |
| Enterprise Scale | 0.073 | 0.030 | 0.014 | 0.056 | 0.016 | 0.014 | 0.008 | 0.005 |
| Founding Time | 0.041 | 0-.003 | -0.020 | 0.018 | -0.022 | -0.025 | -0.016 | -0.021 |
| Development Stage | -0.150 | -0.106 | -0.035 | -0.090 | -0.049 | -0.008 | 0.008 | -0.010 |
| **Independent Variable** | | | | | | | | |
| Knowledge Search Breadth | | 0.714*** | | | 0.570*** | | 0.394** | |
| Knowledge Search Depth | | | 0.799*** | | | 0.659*** | | 0.210** |
| **Mediating Variables** | | | | | | | | |
| Knowledge Integration | | | | | | | 0.371** | 0.450*** |
| $R^2$ | 0.234 | 0.441 | 0.597 | 0.15 | 0.284 | 0.340 | 0.345 | 0.360 |
| $F$ | 4.029 | 66.848*** | 125.515*** | 1.782 | 33.569*** | 43.727*** | 44.670*** | 38.084*** |

enterprise breakthrough innovation. Under the control of the influencing variables of enterprise size, founding time, and development stage, it can be seen from the analysis results that there is a significant positive relationship between knowledge search breadth and enterprise breakthrough innovation ($\beta$ = 0570, p<0.001) and the relationship between knowledge search depth and enterprise breakthrough innovations is also positive ($\beta$ = 0.659, p<0.001). Also, we can find that the knowledge search depth has a greater impact on enterprise breakthrough innovation. The above analysis results support Hypothesis H1a and H1b.

**Mediating effect test.** According to the test conditions of the Mediating effect, the test of the independent variable on the dependent variable has been completed in the main effect test stage. Under the condition of controlling the influence variables of enterprise size, founding time and development stage, it can be seen from Model 2 that there is a significant positive relationship between knowledge search breadth and knowledge integration($\beta$ = 0.714, p<0.001). Also, according to Model 3, we can see the relationship between knowledge search depth and knowledge integration is significant positive($\beta$ = 0.799, p<0.001). After adding the intermediary variables on the basis of Model 5 and Model 6, the positive impact of knowledge search width on enterprise breakthrough innovation($\beta$ = 0.394, p<0.01) in Model 7 is reduced, and the positive effect of knowledge integration on enterprise breakthrough innovation($\beta$ = 0.371, p<0.01) is significant. From the above analysis, we can see that knowledge integration plays a part of the mediating role in the relationship between knowledge search breadth and enterprise breakthrough innovation. In the Model 8, the positive impact of knowledge search depth on enterprise breakthrough innovation ($\beta$ = 0.210, p<0.01) is reduced, and the positive effect of knowledge integration on enterprise breakthrough innovation ($\beta$ = 0.450, p<0.001) is significant, thus it can be seen that knowledge integration plays a part of the mediating effect in the relationship between knowledge search depth and enterprise breakthrough innovation, while the mediating effect is stronger, which could support Hypothesis H2a and H2b. As shown in Table 7, the Bootstrapping method is used to further test the robustness of the mediating effect of knowledge integration, and the 95% confidence interval does not contain 0, which indicates that the mediating effect of knowledge integration is significant and further supports H2a and H2b by data.

**Moderating effect test.** In order to test the moderating effect of the innovation ecosystem network characteristics on the relationship between knowledge search and knowledge integration, we generate the interaction terms of "knowledge search width×network size" and "knowledge search depth×network connection strength" based on the proposed hypothesis,

**Table 7. Bootstrapping mediating effect test of knowledge integration.**

| Variable | Proportion of Mediating Effect | 95%Confidence Interval | | Conclusion |
|---|---|---|---|---|
| | | **BootLLCI** | **BootULCI** | |
| Knowledge Search Breadth—Knowledge Integration—Enterprise Breakthrough Innovation | 46.473% | 0.129 | 0.340 | Partial intermediary |
| Knowledge Search Depth—Knowledge Integration—Enterprise Breakthrough Innovation | 54.560% | 0.215 | 0.457 | Partial intermediary |

Note: The sample size is 5000, BootLLCI and BootULCI refer to the lower limit and upper limit of 95% sampling interval respectively.

and use knowledge integration as the dependent variable for hierarchical regression analysis. As shown in Table 8, in the case of controlling the influencing variables of enterprise size, establishment time, and development stage, Model 1, Model 3, and Model 5 examine the moderating effect of innovation ecosystem network size in the knowledge search breadth and knowledge integration. We can see that knowledge search width×network size" has a significant positive impact on knowledge integration($\beta = 0.353$, p<0.01), which means that the larger the innovation ecosystem network size, the stronger the positive impact of knowledge search width on knowledge integration. Also, Hypothesis H3a is supported. Model 2, Model 4, and Model 6 test the moderating effect of innovation ecosystem network connection strength in the relationship between knowledge search depth and knowledge integration. The results show that "knowledge search depth×network connection strength" has a significant positive impact on knowledge integration ($\beta = 0.489$, p<0.01). The closer the innovation ecosystem network connection strength is, the stronger the positive relationship between knowledge search depth and knowledge integration. Finally, Hypothesis H3b is supported.

In order to further explain the above empirical results, we draw graphs of the moderating effects of different levels of innovation ecosystem network characteristics on the relationship between knowledge search and knowledge integration, as shown in Fig 2.

**Moderated mediation test.** We use the PROCESS plug-in in SPSS25.0 software and the Bootstrap method to further test the moderated mediation model, and add or subtract the

**Table 8. The moderating effect test of innovation ecosystem network characteristics.**

| Variable | Knowledge Integration | | | | | |
|---|---|---|---|---|---|---|
| | **Model 1** | **Model 2** | **Model 3** | **Model 4** | **Model 5** | **Model 6** |
| **Control Variable** | | | | | | |
| Enterprise Scale | 0.030 | 0.014 | 0.011 | 0.007 | 0.011 | 0.007 |
| Founding Time | -0.003 | -0.020 | -0.003 | -0.024 | -0.002 | -0.024 |
| Development Stage | -0.106* | -0.035 | -0.087 | -0.020 | -0.087 | -0.024 |
| **Independent Variable** | | | | | | |
| Knowledge Search Breadth | 0.714** | | 0.558** | | 0.557** | |
| Knowledge Search Depth | | 0.799** | | 0.724** | | 0.725** |
| **Moderating Variable** | | | | | | |
| Network Size | | | 0.342** | | 0.343** | |
| Network Connection Strength | | | | 0.165** | | 0.162* |
| **Interaction term** | | | | | | |
| Knowledge Search Breadth× Network Size | | | | | 0.353** | |
| Knowledge Search Depth×Network Connection Strength | | | | | | 0.489** |
| $R^2$ | 0.441 | 0.597 | 0.479 | 0.606 | 0.479 | 0.607 |
| $F$ | 66.834** | 125.576** | 62.211** | 104.021** | 51.776** | 86.988** |

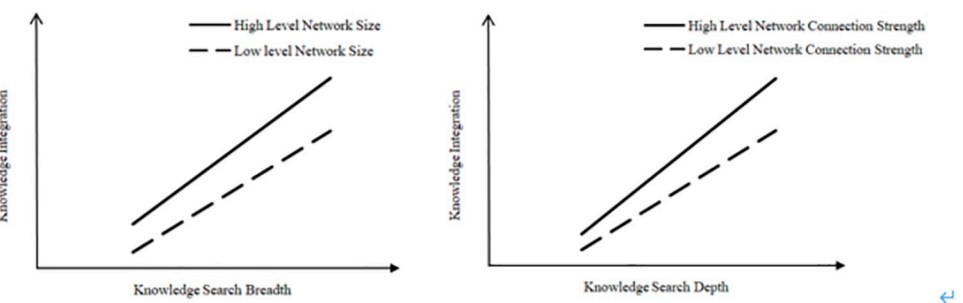

**Fig 2. The moderating effect of network characteristics of innovation ecosystem.**

average value of the mediation variable by one standard deviation to distinguish it in three levels. The sample has been repeatedly drawn 5000 times and the confidence interval is 95% (as shown in Table 9). For high-level network size, knowledge search breadth has a significant impact on enterprise breakthrough innovations through knowledge integration($\beta$ = 0.382, at a 95% confidence level, confidence interval [0.105, 0.330], excluding 0). For the low-level network size, the impact of knowledge search width on enterprise breakthrough innovation through knowledge integration is not significant ($\beta$ = 0.127, at the 95% confidence level, the confidence interval is [0.004, 0.316], although 0 is included, the lower limit is close to 0), which means that as the innovation ecosystem network size becomes larger, the mediating role of knowledge integration on the relationship between knowledge search breadth and enterprise breakthrough innovation becomes stronger. Therefore, the Hypothesis H4a is supported. For the high-level network connection strength, knowledge search depth has a significant impact on the enterprise breakthrough innovation through knowledge integration($\beta$ = 0.422, at the 95% confidence level, confidence interval [0.123, 0.696], excluding 0). In terms of low-level network size, knowledge search depth has no significant impact on the relationship between knowledge integration and enterprise breakthrough innovation($\beta$ = 0.174, at the 95% confidence level, confidence interval [-0.211, 0.411], including 0). The results indicates that as the innovation ecosystem network connection strength increases, the mediating role of knowledge integration on the relationship between knowledge search depth and enterprise breakthrough innovation becomes stronger, which could support Hypothesis H4b.

## Conclusions and discussion

### Research conclusion

This paper takes the innovation ecosystem as the boundary of the research paradigm. Based on the innovation ecosystem theory, knowledge management theory and enterprise innovation

**Table 9. The moderating mediating effect test.**

| Moderating Variable | Conditional Indirect Effect | | 95%Confidence Interval | |
|---|---|---|---|---|
| | Effect Coefficient | Standard Error | BootLLCI | BootULCI |
| Knowledge Search Breadth—Knowledge Integration—Enterprise Breakthrough Innovation | | | | |
| Low level Network Size(2.918) | 0.127 | 0.055 | 0.004 | 0.316 |
| Average level Network Size(3.462) | 0.207 | 0.054 | 0.106 | 0.317 |
| High Level Network Size(4.006) | 0.382 | 0.058 | 0.105 | 0.330 |
| Knowledge Search Depth—Knowledge Integration—Enterprise Breakthrough Innovation | | | | |
| Low Level Network Connection Strength(2.750) | 0.174 | 0.062 | -0.211 | 0.412 |
| Average Level Network Connection Strength(3.518) | 0.330 | 0.061 | 0.207 | 0.447 |
| High Level Network Connection Strength(4.286) | 0.422 | 0.064 | 0.123 | 0.696 |

theory, we construct the moderated mediation model of the enterprise knowledge search, knowledge integration and breakthrough innovation under the innovation ecosystem network characteristics, and put forward research hypotheses. Through the investigation and analysis of 344 technology enterprises in Beijing-Tianjin-Hebei region, the following research conclusions are drawn.

Firstly, in the innovation ecosystem, the knowledge search breadth and knowledge search depth have a positive effect on enterprise breakthrough innovation. This is because through external knowledge search, enterprise can continuously obtain the knowledge resources of other innovation entities in the innovation ecosystem, such as universities and scientific research institutions, etc, as well as the potential value of customers or suppliers and the cutting-edge knowledge in this field, which could provide the latest knowledge resources and information for enterprise breakthrough innovation.

Secondly, knowledge integration plays a part of moderating role in the relationship among knowledge search breadth and knowledge search depth and enterprise breakthrough innovation, that is, knowledge integration promotes enterprise innovation. At the same time, it explains that in order to improve performance, external knowledge must be effectively integrated and utilized within the enterprise. From the data of mediating effect, the mediation effect of knowledge integration in the relationship between knowledge search depth and enterprise breakthrough innovation is stronger.

Thirdly, the innovation ecosystem network characteristics have a moderating effect on knowledge search and knowledge integration. Based on the research hypothesis, we further test the moderated mediating effect and find that the positive moderating effect of the innovation ecosystem network characteristics on the mediating effect relationship between knowledge search breadth and enterprise breakthrough innovation. The larger the size, the stronger the network connection, the stronger the mediating role of knowledge integration.

As the technology enterprises are mainly engaged in the research and development of high-tech products, they have similar characteristics such as paying attention to innovation and R&D investment. Therefore, the above research conclusions have certain universal value for the management of technology enterprises in other regions.

## Management enlightenment

The conclusions of this study have important implications for enterprise management practices, including.

Firstly, actively use knowledge resources in the innovation ecosystem to promote enterprise knowledge innovation. This paper verifies that the network size and network connection strength play a positive mediation role in the relationship between knowledge search and knowledge integration. In the innovation ecosystem, enterprise should continuously absorb and integrate external knowledge for their own use. While carrying out independent research and development, enterprise should establish active and stable cooperative relations with universities and scientific research institutions in the region, expand the scope of cooperation, strengthen communication and exchanges with regional innovation entities, and learn from each other's strengths to master more knowledge and technology that can enhance knowledge innovation capabilities.

Secondly, clarify the different promotion effects of knowledge search breadth and knowledge search depth on enterprise innovation. When searching for knowledge, we should not only pay attention to the knowledge search breadth, but also the knowledge search depth. So we can enrich the stock of heterogeneous knowledge resources by increasing the knowledge search breadth and obtain the cutting-edge knowledge by strengthening the knowledge search depth.

Thirdly, continuously improve the company's knowledge integration capabilities. A large amount of heterogeneous knowledge is stored in external knowledge sources. So, enterprise must clarify the criteria for identifying effective knowledge according to their innovation needs, which is the starting point of enterprise knowledge integration. After clearing the identification criteria, the enterprise should improve the ability to integrate and utilize internal knowledge, classified manage and effectively integrate the internal knowledge of the organization with the external resources which have been identified and acquired.

## Research deficiencies and prospects

Although this study uses a variety of methods from data collection to statistical analysis to improve the reliability and validity of the sample, and the research hypothesis has basically been verified, there are still some deficiencies that need to be improved due to conditions.

Firstly, although the research conclusion has a certain universal value for the management of technology enterprises in other regions, due to the relatively small sample size, the future research can expand the research area to increase the sample size, so as to further verify the research conclusions of universality.

Secondly, the current measurement of the innovation ecosystem is still in the exploratory stage, and an authoritative measurement questionnaire has not been formed. So, it is necessary to further develop a scientific measurement scale.

Thirdly, this article takes the enterprise size, he establishment time and the development stage as the control variables. However, Other external environmental factors in the enterprise development may also affect the enterprise breakthrough innovation. Therefore, future research could adjust the control variables to test.

Fourthly, due to the limitations of the research conditions, this study selects cross-sectional sample data of enterprise to test the hypothesis. There may be a "time lag effect" in the enterprise breakthrough innovation capabilities. In the future, it is necessary to use longitudinal data to add the time factor to fully test the causal relationship between knowledge search and enterprise breakthrough innovation.

## Supporting information

**S1 File. Questionnaire-Chinese original edition.**
(DOCX)

**S2 File. Questionnaire-English copy edition.**
(DOCX)

**S3 File. Data-Chinese original edition.**
(XLSX)

**S4 File. Data-English copy edition.**
(XLSX)

**S5 File. The description of questionnaire variables.**
(DOCX)

## Acknowledgments

We are grateful to all the participants for their contributions to this study. We would also like to thank the editors and reviewers for their suggestions.

## Author Contributions

**Data curation:** Yanli Zhang, Long Xu.

**Methodology:** Yanli Zhang.

**Writing – original draft:** Yanli Zhang, Dantong Wang, Long Xu.

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
