## [Decision Letter · Decision Letter 0]

18 Oct 2021

PONE-D-21-27668Knowledge Search, Knowledge Integration and Enterprise Breakthrough Innovation under the Characteristics of Innovation Ecosystem Network

——A Moderated Mediation ModelPLOS ONE

Dear Dr.  王,

Thank you for submitting your manuscript to PLOS ONE. After careful consideration, we feel that it has merit but does not fully meet PLOS ONE’s publication criteria as it currently stands. Therefore, we invite you to submit a revised version of the manuscript that addresses the points raised during the review process.

Please consider all comments of all reviewers

Please submit your revised manuscript by Dec 02 2021 11:59PM. If you will need more time than this to complete your revisions, please reply to this message or contact the journal office at plosone@plos.org. Please include the following items when submitting your revised manuscript:A rebuttal letter that responds to each point raised by the academic editor and reviewer(s). You should upload this letter as a separate file labeled 'Response to Reviewers'.A marked-up copy of your manuscript that highlights changes made to the original version. You should upload this as a separate file labeled 'Revised Manuscript with Track Changes'.An unmarked version of your revised paper without tracked changes. You should upload this as a separate file labeled 'Manuscript'.

We look forward to receiving your revised manuscript.

Kind regards,

Ahmed Mancy Mosa, Ph.D.

Academic Editor

PLOS ONE

Journal Requirements:

“We are grateful to all the participants for their contributions to this study. We would also like to thank the supporters from the Ministry of Education of the People's Republic of China's Youth Foundation for Humanities and Social Sciences Research Project (Project No. 19YJC630220) and S&T Program of Hebei (Project No. 21557652D).”

We note that you have provided additional information within the Acknowledgements Section. Please note that funding information should not appear in the Acknowledgments section or other areas of your manuscript. We will only publish funding information present in the Funding Statement section of the online submission form.

 “Zhang YL, 19YJC630220, the Ministry of Education of the People's Republic of China's Youth Foundation for Humanities and Social Sciences Research Project, https://www.sinoss.net

Zhang YL, 21557652D, S&T Program of Hebei, https://kjt.hebei.gov.cn/www/index_ssl/index.html

The sponsors or funders did not play any role in the study design, data collection and analysis, decision to publish, or preparation of the manuscript”

Reviewers' comments:

Reviewer's Responses to Questions

**Comments to the Author**

1. Is the manuscript technically sound, and do the data support the conclusions?

Reviewer #1: Partly

Reviewer #2: Yes

2. Has the statistical analysis been performed appropriately and rigorously? 

Reviewer #1: Yes

Reviewer #2: Yes

3. Have the authors made all data underlying the findings in their manuscript fully available?

Reviewer #1: No

Reviewer #2: No

4. Is the manuscript presented in an intelligible fashion and written in standard English?

Reviewer #1: Yes

Reviewer #2: Yes

5. Review Comments to the Author

Reviewer #1: 1. General comments

The authors’ focus on the role of the innovation ecosystem in enterprise breakthrough innovation is interesting.

However, I have some concerns as described below.

2. Specific comments

a) Major

As for the variables used, it is indicated that they were obtained from the responses to the questionnaire survey. However, there is no specific mention of what kind of questions were actually used. Furthermore, some of the papers referred to in the design of the questions are not in English, making it difficult for other researchers to follow up on this study. Although data such as the Cronbach's a coefficient values are presented, it is not enough to determine whether the data of the responses to the questions are truly representative of each variable. However, this problem could be solved by either including the questionnaire questions in the text or submitting the actual questionnaire items used as an appendix.

b) Minor

In this study, the companies surveyed were limited to a specific country or region. Since corporate activities can be influenced by the characteristics of a country or region, it may be difficult to draw general conclusions from the results of this study. In order to solve this problem, for example, the title could be modified and limited by stating the country and/or region.

Reviewer #2: The work is sound and it provides evidence about how enterprises can perform breakthrough innovation. To assess the role of knowledge integration in achieving breakthrough innovations, the authors rely on how knowledge is search (considering breadth and depth) and how the enterprise is embedded in the innovation ecosystem (network size and cohesion).

The work should be improved by clarifying how the innovation ecosystem is constructed and assessed. The authors may consider previous works related to Social Network Analysis applied to Innovation Systems, and specify how the networks are constructed and which cohesion and centrality metrics are used. At least, a reference to Social network analysis applied to Innovation systems, including cohesion and centrality concepts should be made.

Finally, some reflections related to further research needed to ensure a proper extrapolation of the results to other sectors or countries will be valuable.

6. PLOS authors have the option to publish the peer review history of their article (what does this mean?). If published, this will include your full peer review and any attached files.

Reviewer #1: No

Reviewer #2: No

---

## [Author Response · Author response to Decision Letter 0]

23 Nov 2021

We have revised the manuscript according to the suggestions of reviewers. Please see the "Response to Reviewers" for details.

---

## [Decision Letter · Decision Letter 1]

6 Dec 2021

Knowledge search, knowledge integration and enterprise breakthrough innovation under the characteristics of innovation ecosystem network——the empirical evidence from enterprises in Beijing-Tianjin-Hebei region

PONE-D-21-27668R1

Dear Dr.   ,

We’re pleased to inform you that your manuscript has been judged scientifically suitable for publication and will be formally accepted for publication once it meets all outstanding technical requirements.

Kind regards,

Ahmed Mancy Mosa, Ph.D.

Academic Editor

PLOS ONE

Additional Editor Comments (optional):

Reviewers' comments:

Reviewer's Responses to Questions

**Comments to the Author**

1. If the authors have adequately addressed your comments raised in a previous round of review and you feel that this manuscript is now acceptable for publication, you may indicate that here to bypass the “Comments to the Author” section, enter your conflict of interest statement in the “Confidential to Editor” section, and submit your "Accept" recommendation.

Reviewer #1: All comments have been addressed

2. Is the manuscript technically sound, and do the data support the conclusions?

Reviewer #1: Yes

3. Has the statistical analysis been performed appropriately and rigorously? 

Reviewer #1: Yes

4. Have the authors made all data underlying the findings in their manuscript fully available?

Reviewer #1: Yes

5. Is the manuscript presented in an intelligible fashion and written in standard English?

Reviewer #1: Yes

6. Review Comments to the Author

Reviewer #1: (No Response)

7. PLOS authors have the option to publish the peer review history of their article (what does this mean?). If published, this will include your full peer review and any attached files.

Reviewer #1: No

---

## [Editor Report · Acceptance letter]

10 Dec 2021

PONE-D-21-27668R1 

Knowledge search, knowledge integration and enterprise breakthrough innovation under the characteristics of innovation ecosystem network
——the empirical evidence from enterprises in Beijing-Tianjin-Hebei region 

Dear Dr. Wang:

I'm pleased to inform you that your manuscript has been deemed suitable for publication in PLOS ONE. Congratulations! Your manuscript is now with our production department. 

Kind regards, 

on behalf of

Dr. Ahmed Mancy Mosa 

Academic Editor

PLOS ONE